# Uncovering Hidden Diversity: Three New Species of the *Keratella* Genus (Rotifera, Monogononta, Brachionidae) of High Altitude Water Systems from Central Mexico †



Alma E. García-Morales [1],*, Omar Domínguez-Domínguez [2] and Manuel Elías-Gutiérrez [1]

1 Laboratorio de Zooplancton, El Colegio de la Frontera Sur, Av. Centenario km 5.5, Chetumal 77014, Quintana Roo, Mexico; melias@ecosur.mx

2 Laboratorio de Biología Acuática, Facultad de Biología, Universidad Michoacana de San Nicolás de Hidalgo, Ciudad Universitaria, Morelia 58000, Michoacan, Mexico; goodeido@yahoo.com.mx

* Correspondence: aegarcia@ecosur.mx; Tel.: +52-(983)-835-0440

† urn:lsid:zoobank.org:act:7F194E23-7A36-46C4-90D9-EA38EB5F4380;urn:lsid:zoobank.org:act:FD30C7A8-C63E-435D-A1C6-99003007789B;urn:lsid:zoobank.org:act:DCA4000E-4B2B-4221-A15A-8CF15EF69B30.

**Abstract:** The correct identification of species is an essential step before any study on biodiversity, ecology or genetics. *Keratella* is a genus with a predominantly temperate distribution and with several species being endemics or restricted geographically. Its diversity may be underestimated considering the confusing taxonomy of species complexes such as *K. cochlearis*. In this study, we examined genetic diversity and morphology among some *Keratella* populations from Mexico in order to determine if these populations represent different species. We analyzed a dataset of previously published and newly generated sequences of the mitochondrial COI gene and the nuclear ITS1 marker. We conducted phylogenetic analyses and applied three methods of species delimitation (ABGD, PTP and GMYC) to identify evolutionary significant units (ESUs) equivalent to species. Morphological analyses were conducted through scanning electron microscope (SEM) and morphometry under a compound microscope. In the present study, three new species *Keratella cuitzeiensis* sp. nov., *Keratella huapanguensis* sp. nov., and *Keratella albertae* sp. nov., are formally described. These species were collected in high-altitude water bodies located in the Central Plateau of Mexico. Combining DNA results through COI and ITS1 molecular markers and morphology it was possible to confirm the identity of the new species.

**Keywords:** integrative taxonomy; biodiversity; DNA taxonomy; ABGD; morphology; genetic entities





## 1. Introduction

Species are the fundamental unit of biodiversity; therefore, any biodiversity analyses, as well as genetic, physiological and ecological studies, rely on the proper delimitation and identification of species [1]. However, estimates of species richness are often hampered by the presence of cryptic species, which are groups of species that are not confidently distinguishable based only on morphology [2].

Rotifera is a phylum of microscopic animals (50–2000 μm) that are globally distributed in aquatic ecosystems [3]. They play an essential role in aquatic food webs by transferring energy to higher trophic levels [4,5]. Rotifera harbors a high level of cryptic diversity [6], and this hidden diversity is expected due to the small size of rotifers, the scarcity of rotifer taxonomists that can identify them reliably, the lack of taxonomically relevant morphological features, little or no morphological variation between species, as well as the high level of phenotypic plasticity present in several species [2,7]. Cryptic species complexes have been described for taxa such as *Brachionus plicatilis* (Müller, 1786) [8,9], *B. calyciflorus* Pallas, 1766 [10], *Epiphanes senta* (Müller, 1773) [11], *Polyarthra dolichoptera* (Idelson, 1925) [12], *Keratella cochlearis* (Gosse, 1851) [13,14], *Limnias melicerta* (Weisse, 1848)

and *L. ceratophylli* (Schrank, 1803) [15], and other such as *Platyias quadricornis* (Ehrenberg, 1832) and *Testudinella patina* (Hermann, 1783) [6].

Within Brachionidae, *Keratella* (Bory de St. Vincent, 1822) is the genus with the highest degree of endemicity, mainly in temperate zones (e.g., Palearctic, Nearctic and Australian regions), with some endemics from Neotropical and Oriental regions. The rest of the *Keratella* species are cosmopolitan or widespread in some regions [16]. Currently, there are approximately 53 *Keratella* taxa recognized as valid species [17]. It seems that diversity in *Keratella* is low because of the number of the registered species; nevertheless, diversity within this genus is underestimated due to the presence of cryptic species complexes, for example as *K. cochlearis* s.l. [16]. Morphologically, *Keratella* species bear a stiff lorica, which is split into two plates, one dorsal and a ventral one. This lorica can be rectangular, trapezoidal or ovoid in shape. The dorsal part of the lorica can be smooth or covered by different types of ornamentation such as granules, pustules, spinules or reticulation, whereas the ventral part of the lorica is generally smooth, but may have ornamentation on its anterior part [18].

Besides, the dorsal lorica presents several fields (also named plaques, polygones, panels, facets) with symmetrical and asymmetrical polygonal shapes. The arrangement and shape of these fields of the dorsal plate have taxonomic importance for the identification of the species [18–20]. In *Keratella*, the anterodorsal margin of the lorica presents six curved spines, being the anteromedian pair the most curved inward. Whereas, the posterior margin of the lorica may have two posterolateral spines, a single spine or the posterior spines can be absent [19,20]. The genus can be split roughly into two main groups: (1) the "quadrata" group, which presents a row of median fields over the dorsal plate, and (2) the "cochlearis" group, which has a median ridge over the dorsal plate with the fields arranged on each side of the ridge [18].

On the other hand, *Keratella tropica* (Apstein, 1907) was described from Colombo Lake in Ceylon (now Sri Lanka) by [21]. It is widely distributed in tropical and subtropical regions of the world [22] and has also been reported in Netherlands and Siberia during summer [23]. *K. tropica* is morphologically similar to *K. valga* (Ehrenberg, 1834), but differs from this late by the presence of an additional field on the dorsal lorica: the postero-median remnant [24,25]. In *K. tropica,* the dorsal lorica has a row of five median fields, four of these are hexagonal, and the fifth (the postero-median remnant) is squared [18,19]. In this species, the length of the posterior spines varies widely, as well as the size of the lorica [18]. Sometimes is difficult to observe the dorsal median fields and especially the small remnant, resulting in confusion when examining different specimens within *K. tropica* s.l. Besides the wide variation in the length of the posterior spines and size reported in *K. tropica*, this indicates that some of these variants could be cryptic species.

The development of DNA-based taxonomic tools provides a means to study biodiversity through the analysis of genetic variation in molecular markers to delimit species [26]. Markers such as the mitochondrial cytochrome c oxidase subunit 1 (COI) and the nuclear internal transcribed spacer (ITS) were useful for the study of cryptic speciation, and genetic differentiation in some rotifer taxa [2,27]. In addition, combining molecular, morphological, ecological and crossmating analyses have proved to be a suitable approach to assess cryptic diversity and to delimit species in rotifers [2,26]. This approach falls in the so-called integrative taxonomy [28]. For example, in the well studied *B. plicatilis* complex, morphology and molecular analyses and mating experiments have shown that this taxon contains at least 14 possible cryptic species [2,29]. From these, only *B. paranguensis* Guerrero-Jiménez, Vannucchi, Silva-Briano, Adabache-Ortiz, Rico-Martínez, Roberts, Neilson, Elías-Gutiérrez, 2019 was described [9]. Another study with the widespread *Epiphanes senta* demonstrated that this taxon is a species complex and three new species were formally described based on morphological and genetic evidence [10].

In the present study, we used integrative taxonomy tools to explore the diversity of some *Keratella* populations from seven water bodies of Mexico. Specimens from these populations resemble *K. tropica* morphologically.

## 2. Materials and Methods

### 2.1. Sampling

We collected samples in seven high altitude (>1700 m above sea level, masl) water bodies along the Trans-Mexican Volcanic Belt (TMVB): Santa Teresa dam, Cuitzeo lagoon, Huapango dam, Yuriria dam, Tepatitlan-Yahualica pond, Ignacio Ramirez dam and Timilpan pond. One additional sample was taken near Kohunlich in the lowlands of the Yucatan Peninsula (Figure 1, Table S1). TMVB is a morphotectonic province extended from Gulf of Mexico to Pacific Ocean in central Mexico [30]. It is characterized to be a cold region formed by complex mountains and volcanoes with average annual temperatures varying between 12–22 °C and average annual precipitation between 300–2000 mm depending on the zone [31]. We collected samples using a 50 µm mesh size plankton net. Posteriorly, all samples were sieved to extract all water, and fixed with 96% ethanol. Samples were transported on ice to the laboratory, and stored in a freezer.

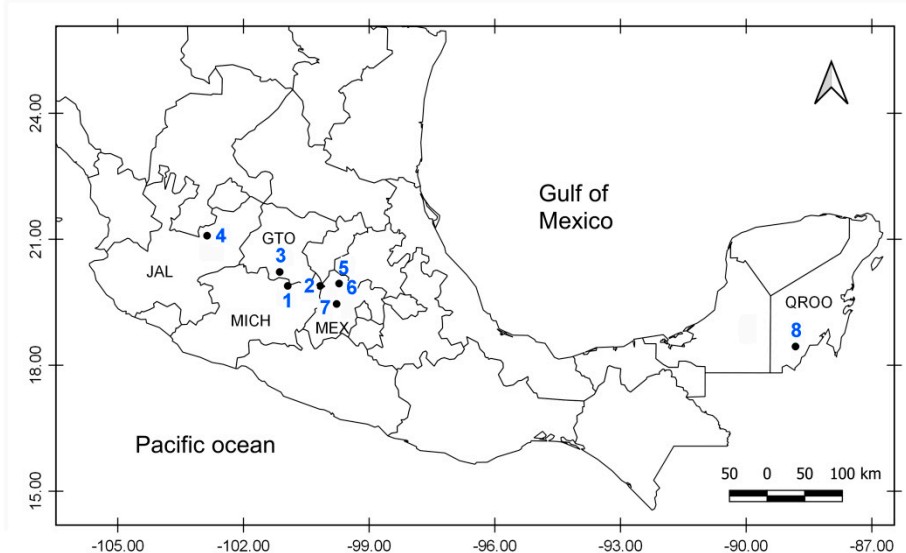

**Figure 1.** Sample locations of the *Keratella* taxa from Mexico. Numbers on the map correspond to the numbers in Table S1. Locations 1 to 5 were collected in this study. Samples for locations 6 to 8 were obtained from [6]. 1 = Cuitzeo lagoon, 2 = Santa Teresa dam, 3 = Yuriria dam, 4 = Tepatitlan-Yahualica pond, 5 = Huapango dam, 6 = Timilpan pond, 7 = Ignacio Ramirez dam, 8 = Aguada Kohunlich. JAL = Jalisco state, GTO = Guanajuato state, MICH = Michoacan state, MEX = Mexico state, QROO = Quintana Roo state.

### 2.2. DNA Extraction and Amplification

We sorted *Keratella* specimens from the samples under a stereomicroscope, rinsed them with distilled water to remove debris, and transferred them into PCR tubes for DNA extraction. We conducted the DNA extraction and PCR amplification of the COI gene according to [6], using the primers LCO1490 and HCO2198 [32]. Additionally, the ITS1 nuclear marker was amplified and the amplification profile is provided in Table S2. For ITS1 we used the primers III 5′-CACACCGCCCGTCGCTACTACCGATTG-3′ and VIII 5′-GTGCGTTCGAAGTGTCGATGATCAA-3′ from [33]. We deposited all sequences from COI and ITS markers in GenBank under accession numbers, COI: OL678378-OL678396 and ITS1: OL664525-OL664561.

### 2.3. Alignment and Phylogenetic Analyses

We downloaded COI (mitochondrial) and ITS1 (nuclear) sequences of *Keratella* taxa from different parts of the world available from GenBank and Barcode of Life Database (BOLD, boldsystems.org) (398 for COI and 157 for ITS1), and included them in this study.

Accession numbers for all the downloaded sequences are available in Table S3. In total, we aligned 417 COI sequences (19 from this study and 398 obtained from GenBank and BOLD), using MEGA 7.0 [34] through ClustalW with default settings. For ITS1, we aligned 194 sequences (37 from this study and 157 from GenBank) through MAFFT v.7 using the Q-INS-I algorithm as the optimal strategy for ribosomal markers [35]. This last aligment was carried out on the MAFFT webserver http://mafft.cbrc.jp/alignment/server/index.html (accessed on 29 March 2021). Both final alignments were subsequently reviewed by us.

First, we ran JMODELTEST v.2.1.1 [36] to identify the model of molecular evolution that best fit the COI (TVM + I + G), and ITS1 (HKY + G) datasets, defined by the Akaike Information Criterion. Posteriorly, all sequences for each dataset were collapsed in haplotypes using DNASP v.5.10 [37]. We used Bayesian inference (BI) and maximum likelihood (ML) analyses to infer phylogenetic relationships among the different *Keratella* samples with mtDNA COI and nuclear ITS1 analyzed separately. We did not carry out concatenated analyses because our sequences of the COI and ITS1 markers were obtained from different individuals, except five individuals from whom we have their COI and ITS1 sequences (JX216635, JX216636, JX216637, JX216638 and JX216639). We conducted The BI and ML analyses through MrBAYES v.3.2.7 [38] and RAXML v.1.5 [39], respectively. The settings for the BI analysis for each molecular dataset were four simultaneous Markov Chain Monte Carlo (MCMC) runs for six and five million generations for mitochondrial and nuclear data, respectively, with trees sampled every 100 generations. We used TRACER v.1.7 [40] to assess convergence between runs and monitor the standard deviation of split frequencies and by using the effective sampling size (ESS) criterion (>200), discarding 25% of generations as burn-in to construct the majority-rule consensus tree. For ML analysis we used a GTR + I + G (mtDNA data), and GTR + G (nuclear data) models and we ran both ML analyses with 5000 bootstrap replications. We used *Plationus patulus* (Müller, 1786) (accession number JX216784 for COI and KC431010 for ITS1) as outgroup for the phylogenetic analyses.

### 2.4. Species Delimitation

We applied three methods of species delimitation for mtDNA and nrDNA datasets and compared the results. Generalized Mixed Yule-Coalescent model (GMYC) [41], uses a maximum likelihood approach to identifying the shift in the branching patterns between species level (Yule model) and population level (Coalescent model) to delimit independently evolving entities. For the GMYC method, we generated ultrametric trees from the two datasets (COI and ITS1) using BEAST v.2.1.3 [42]. The settings comprised a GTR + G + I (for COI) and HKY + G (for ITS1) substitution model, a relaxed lognormal clock, and a birth–death prior [43]. Because of the absence of a molecular clock specific to Rotifera, we used calibration clocks for COI of 1.76% sequence divergence per Myr [44] and 1.2% per Myr for ITS1 [45,46] tested in aquatic invertebrates. We ran the analyses with 100 million MCMC for COI and 70 million MCMC for ITS1, sampling every 1000 generations. We checked the MCMC runs for convergence in TRACER v.1.7 [40]. We combined trees in TREEANNOTATOR 2.1.2 using a maximum credibility tree, with the first 10% discarded as burn-in. We ran the GMYC model through the GMYC webserver, using the single threshold option (http://species.h-its.org/gmyc/) (accessed on 1 April 2021).

We conducted the Poisson Tree Processes method (PTP) [47] to search for evidence of independently evolving entities considered to be species. This method uses a phylogenetic tree as input, optimizing differences in branching events in terms of number of substitutions, and adding support values to that branching events. We used the ML trees (from COI and ITS1) generated in the phylogenetic analyses. We ran both analyses with 500,000 MCMC generations on the PTP webserver http://species.h-its.org/ (accessed on 2 April 2021) and using the two types of PTP: Maximum likelihood approach (PTP-ML) and Bayesian approach (PTP-B). Before running the PTP analyses we discarded the outgroup. We also applied the Automatic Barcode Gap Discovery method (ABGD) for COI and ITS1 markers. This method clusters sequences based on the genetic distances by detecting the gaps

(barcode gap) in the distribution of genetic pairwise distances. Thus, the genetic distance among individuals belonging to the same species is smaller than the distance between individuals from different species [48]. We carried out ABGD analyses through its online webserver https://bioinfo.mnhn.fr/abi/public/abgd/abgdweb.html (accessed on 2 April 2021), using default settings.

For this study purpose, we are going to consider the genetic entities discriminated by all species delimitation methods as evolutionary significant units (ESUs), and we will distinguish between COI-ESUs and ITS1-ESUs accordingly to [49].

### 2.5. Measurements of Specimens and Morphological Analyses

We identified specimens morphologically following the taxonomic keys of [18–20]. Specimens were identified using the features of the lorica, mainly its overall shape, shape of the lorica plates, length of the caudal spines and shape of the median fields. Several specimens from the seven populations sampled in this study were separated under a stereomicroscope and measured on a compound microscope Olympus BX51 at 40× using a micrometer.

Morphometric parameters considered for the study were: TL (total length), LL (lorica length without considering anterior and posterior spines), LW (lorica width), RPS, LPS (right and left posterior spines) and AMS, AIS, ALS (anteromedian, intermediate and lateral spines), following [50] (See Figure 2). Body dimensions are in micrometers. We also observed the five main median fields of the dorsal plate: FMF (frontomedian field), AMF (anteromedian field), MMF (mesomedian field), PMF (posteromedian field) and PMR (posteromedian remnant) (See Figure 2). For these analyses, some *Keratella* specimens were gold-coated to be observed in the scanning electron microscope (SEM) JEOL-JSM6010 located in El Colegio de la Frontera Sur in Chetumal.

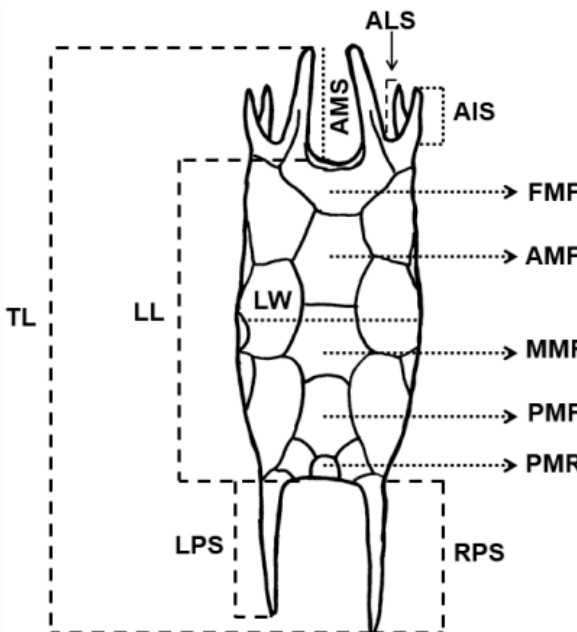

**Figure 2.** Lorica drawing of *Keratella* sp., with measured parameters. Total length (TL), lorica length excluding anterior and posterior spines (LL), lorica width at its widest part (LW), right posterior spine length (RPS), left posterior spine length (LPS), anteromedian dorsal spine length (AMS), anterointermediate dorsal spine length (AIS), anterolateral dorsal spine length (ALS). Frontomedian field (FMF), anteromedian field (AMF), mesomedian field (MMF), posteromedian field (PMF) and posteromedian remnant (PMR).

Our specimens of *Keratella* from the seven sampling sites were compared with the type material of *K. tropica*, which was deposited by [21] in the "Vermes" collection of the Zoological Museum of Berlin with catalog number 10121, in order to determine if our specimens morphologically correspond or not to the *K. tropica* species.

### 2.6. Statistical Analysis of Morphological Measurements

Morphometric measures transformed as the square root of a + ā of adult females were examined with a Principal Component Analysis (PCA) performed with the Multi Variate Statistical Package (MVSP 3.21). We performed a PCA analysis to investigate if the populations examined in this study could be distinguished as separated entities based in morphological measurements. To perform the PCA analysis we measured specimens from the seven sampling sites (see above section).

## 3. Results

### 3.1. DNA Taxonomy

The COI alignment was 580 bp, defining 80 unique haplotypes from 417 sequences; while, ITS1 alignment was 350 bp, with 57 haplotypes from 194 sequences. The trees produced by the BI and ML methods for both markers retrieved the same topology. We present only the ML trees (Figure 3 for COI and Figure 4 for ITS1). For the COI gene, nine well-defined lineages were discriminated, while for the ITS1 marker, four lineages were discriminated (See Figures 3 and 4). We consider two "Keratella tropica" groups in the COI and ITS1 trees, because specimens grouped in these lineages, morphologically resemble the *K. tropica* species (Figures 3 and 4).

In particular, within the "Keratella tropica 1 and 2" lineages for COI gene the three delimitation methods discriminated the same six ESUs (ESU1-ESU6 in Figure 3). ESU1 corresponds to the Ignacio Ramirez and Timilpan populations and were identified as *K*. cf. *morenoi*. ESU2 corresponds to the Cuitzeo population and it is *Keratella cuitzeiensis* sp. nov., ESU3 is from the Huapango population and was named *Keratella huapanguensis* sp. nov. Whereas ESU4 from the Santa Teresa population and ESU5 from the Kohunlich population, both considered as *Keratella albertae* sp. nov. ESU6 include the Yuriria and Tepatitlan-Yahualica populations herein considered here as *K. tropica* s. str., (Figure 3). For the ITS1 marker, within the "Keratella tropica I and II" lineages the three species delimitation tests delimited the same four ESUs, and we called these ESUI to ESUIV (Figure 4). With this marker individuals from the Huapango population (ESU3 with COI) correspond to the ESUI, and individuals from China and Mexico (ESU6 with COI) were nested forming the ESUII. Cuitzeo, Ignacio Ramirez and Timilpan populations (ESU1 and ESU2 with COI) were nested together within the ESUIII, whereas individuals from Santa Teresa and Kohunlich populations (ESU4 and ESU5 with COI) were nested together within the ESUIV (Figure 4). We must clarify that ITS1 sequences are not from the same individuals, except for some specimens (See Methods Section).

The uncorrected *p* distances within the six COI-ESUs (ESU1 to ESU6) ranged from 0 to 0.9% (Table S4), whereas distances between these ESUs ranged from 4 to 20% (Table S4). In the four ITS1-ESUs (ESUI to ESUIV), the uncorrected *p* distances within these ESUs ranged from 0 to 0.5% (Table S5); whereas distances between these ESUs ranged from 3.8 to 10.8% (Table S5). Most of the COI and ITS1 ESUs were formed by only a single haplotype (not a singleton) which were present in a single location, except ESU6 which was formed by four haplotypes as well ESUII and ESUIII with 12 and two haplotypes respectively.

According to the DNA results, and following a conservative approach, we propose the existence of three new *Keratella* species based on the analysis of both markers, supported also with the morphological analyses (see below). These three species are clearly genetically different from the *Keratella tropica* species.

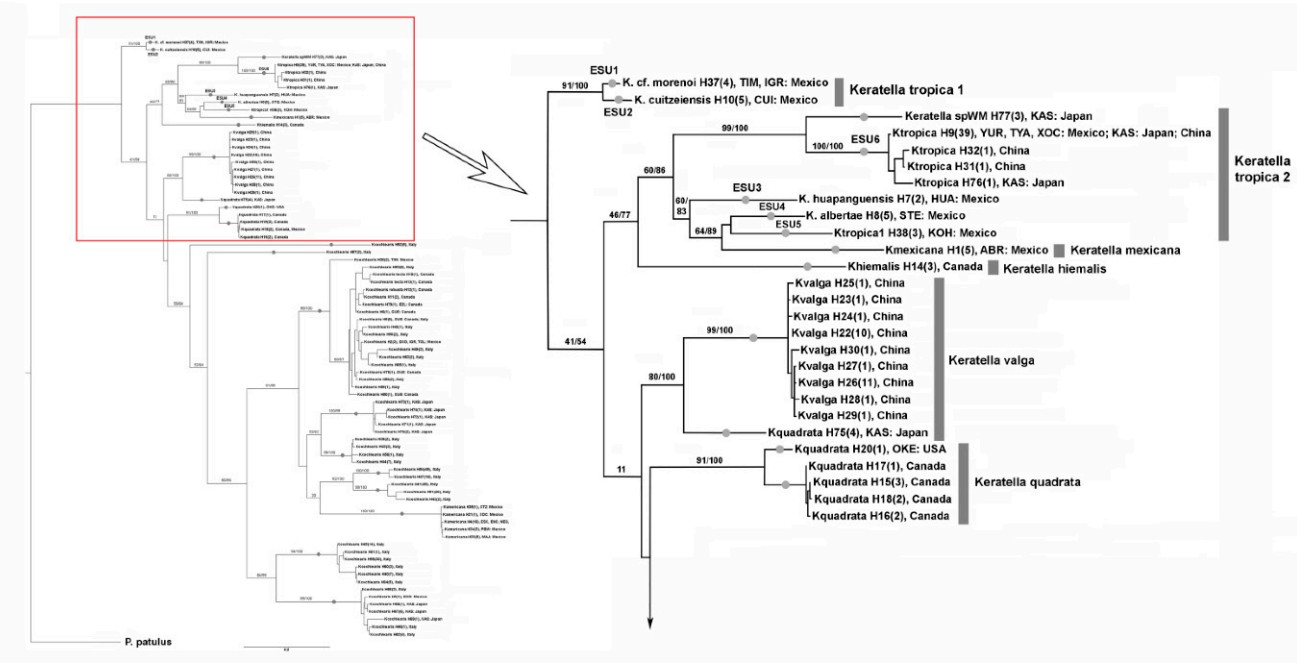

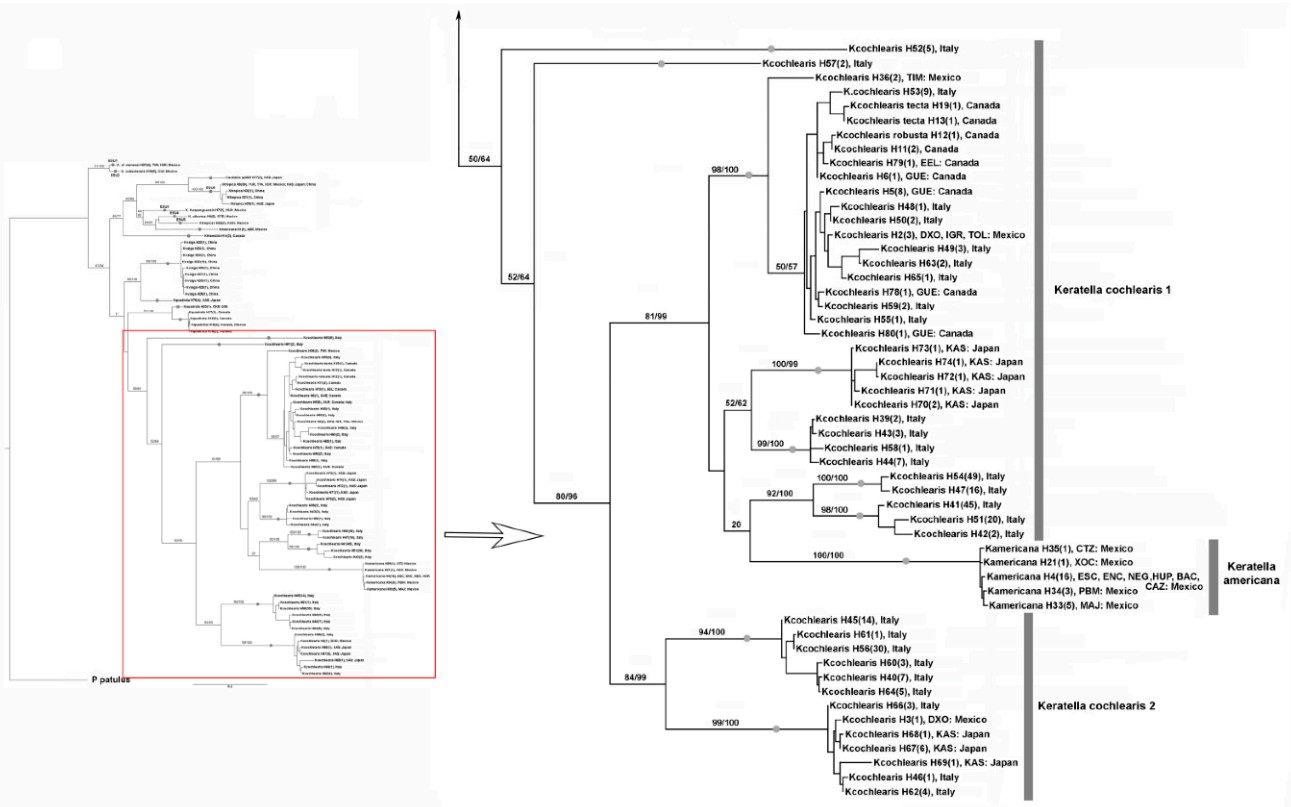

**Figure 3.** Maximum likelihood phylogram of the COI gene showing the relationship of the *Keratella* taxa. Haplotypes are accompanied by the number of individuals displaying that particular haplotype within parentheses and by the acronym of the water body or country in which they were isolated (when the information is available). Numbers on major branches are the percentages of branch support in the Maximum likelihood (bootstrap) and Bayesian (posterior probability) analyses respectively. Dark bars indicate that haplotypes are part of a *Keratella* group. Dark circles over branches indicate a putative species delimited by all the species delimitation methods.

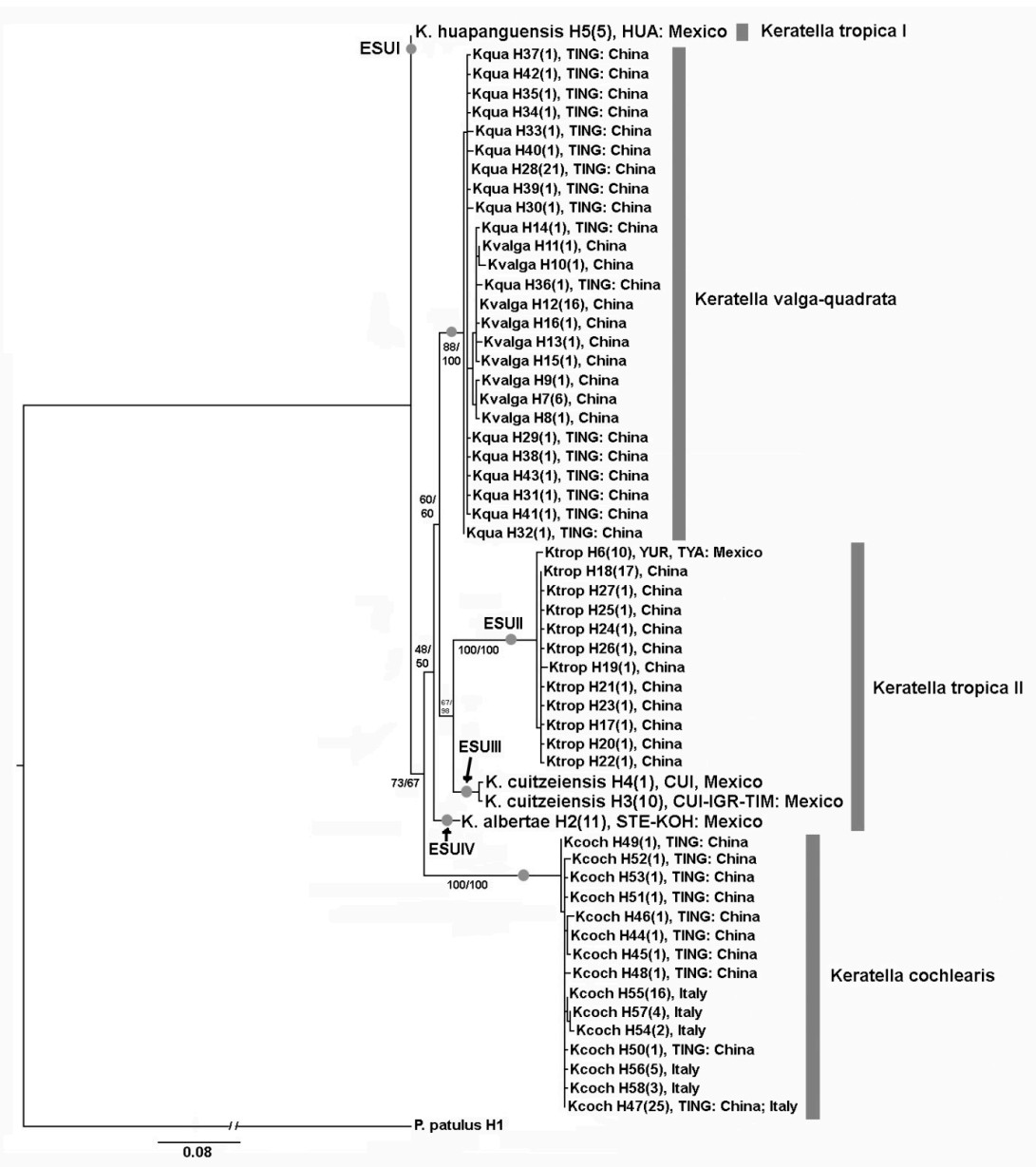

**Figure 4.** Maximum likelihood phylogram of the ITS1 nuclear marker showing the relationship of the *Keratella* taxa. Haplotypes are accompanied by the number of individuals displaying that particular haplotype within parentheses and by the acronym of the water body or country in which they were isolated (when the information is available). Numbers on major branches are the percentages of branch support in the Maximum likelihood (bootstrap) and Bayesian (posterior probability) analyses respectively. Dark bars indicate that haplotypes are part of a *Keratella* group. Dark circles over branches indicate a putative species delimited by all the species delimitation methods.

### 3.2. Statistical Analysis of Morphometric Measurements

We obtained lorica measurements from 84 individuals of *Keratella* taxa from seven populations of Mexico (Tables 1 and S6). From the biplot, a gradient was observed on axis 1, representing the morphometric features in the specimens. Axis 1 explained 98% of the variability. Populations formed a gradient without a well-defined grouping, but a subtle separation can be observed in some populations. Specimens from Ignacio Ramirez and Cuitzeo populations (ESU1 and ESU2 with COI, ESUIII with ITS1) formed a group (Figure 5, see Table 1). Specimens from the Huapango population (ESU3 with COI and ESUI with ITS1) formed another group a little more separated from the other populations

(Figure 5, see Table 1). Specimens from Santa Teresa and Kohunlich populations (ESU4 and ESU5 with COI and ESUIV with ITS1) are mixed and together form a group. The specimens in these two populations are similar in size and morphometric measurements, besides several specimens from the Yuriria population (COI-ESU6 and ITS1-ESUII) are grouped with these previous populations because of their similar size (Figure 5, Table 1). Finally, specimens from the Tepatitlan-Yahualica population (COI-ESU6 and ITS1-ESUII) formed another group separated from the other populations because specimens are small in size compared with the other populations (Figure 5, Table 1). Specimens from Yuriria are separated from specimens from Tepatitlan-Yahualica despite belonging to the same ESU, due to the great morphometric variation in this ESU (which corresponds to *K. tropica* s. str.).

**Table 1.** Length measurements of main lorica features based on 84 specimens of *Keratella* taxa. Measurements were obtained by population, which match with the COI-ESUs. Total length (TL), lorica length excluding anterior and posterior spines (LL), lorica width at its widest part (LW), right posterior spine (RPS), left posterior spine (LPS), anteromedian dorsal spine (AMS), anterointermediate dorsal spine (AIS), anterolateral dorsal spine (ALS). Number of individuals measured is given between brackets.

| | TL | LL | LW | RPS | LPS | AMS | AIS | ALS |
|---|---|---|---|---|---|---|---|---|
| **COI-ESU1** | | | | | | | | |
| Mean | 312.9 | 149.6 | 84.8 | 130.4 | 49.3 | 32.8 | 25.9 | 20.4 |
| Median | 314 | 150 | 86 | 131 | 51 | 32 | 25 | 20 |
| Min | 297 | 142 | 77 | 115 | 30 | 25 | 25 | 17 |
| Max | 322 | 157 | 87 | 147 | 57 | 37 | 30 | 25 |
| Population = Ignacio Ramirez (12) | | | | | | | | |
| **COI-ESU2** | | | | | | | | |
| Mean | 293.4 | 132.3 | 75.4 | 134.8 | 49.7 | 26.1 | 17.9 | 15.8 |
| Median | 301 | 132 | 75 | 142 | 50 | 26 | 18 | 16 |
| Min | 260 | 124 | 74 | 100 | 42 | 26 | 17 | 15 |
| Max | 314 | 140 | 80 | 152 | 62 | 27 | 18 | 16 |
| Population = Cuitzeo (15) | | | | | | | | |
| **COI-ESU3** | | | | | | | | |
| Mean | 201.3 | 113.3 | 65.9 | 56.1 | 45.9 | 31.8 | 19.6 | 17.6 |
| Median | 202 | 114 | 66 | 56 | 45 | 32 | 20 | 18 |
| Min | 194 | 108 | 64 | 52 | 40 | 30 | 19 | 17 |
| Max | 208 | 117 | 72 | 60 | 58 | 32 | 20 | 18 |
| Population = Huapango (15) | | | | | | | | |
| **COI-ESU4** | | | | | | | | |
| Mean | 265.6 | 128.4 | 80.6 | 101.4 | 63 | 35.6 | 21.8 | 19.8 |
| Median | 264 | 128 | 80 | 100 | 67 | 36 | 22 | 20 |
| Min | 260 | 126 | 74 | 96 | 48 | 34 | 21 | 19 |
| Max | 274 | 132 | 84 | 106 | 70 | 36 | 22 | 20 |
| Population = Santa Teresa (10) | | | | | | | | |
| **COI-ESU5** | | | | | | | | |
| Mean | 257.7 | 109.6 | 74.4 | 104.7 | 58.1 | 43.3 | 22.5 | 20.2 |
| Median | 259 | 107 | 75 | 105 | 57 | 43.5 | 22 | 20 |
| Min | 240 | 105 | 67 | 95 | 52 | 40 | 20 | 17 |
| Max | 275 | 125 | 77 | 122 | 65 | 47 | 25 | 22 |
| Population = Kohunlich (12) | | | | | | | | |
| **COI-ESU6** | | | | | | | | |
| Mean | 215.8 | 103.6 | 61.4 | 84.7 | 31.9 | 27.3 | 20.4 | 19.4 |
| Median | 206 | 99 | 63 | 80 | 30 | 26 | 20 | 20 |
| Min | 168 | 90 | 48 | 54 | 8 | 20 | 18 | 18 |
| Max | 264 | 120 | 70 | 118 | 50 | 36 | 24 | 20 |
| Population = Yuriria (10), Tepatitlan (10) | | | | | | | | |

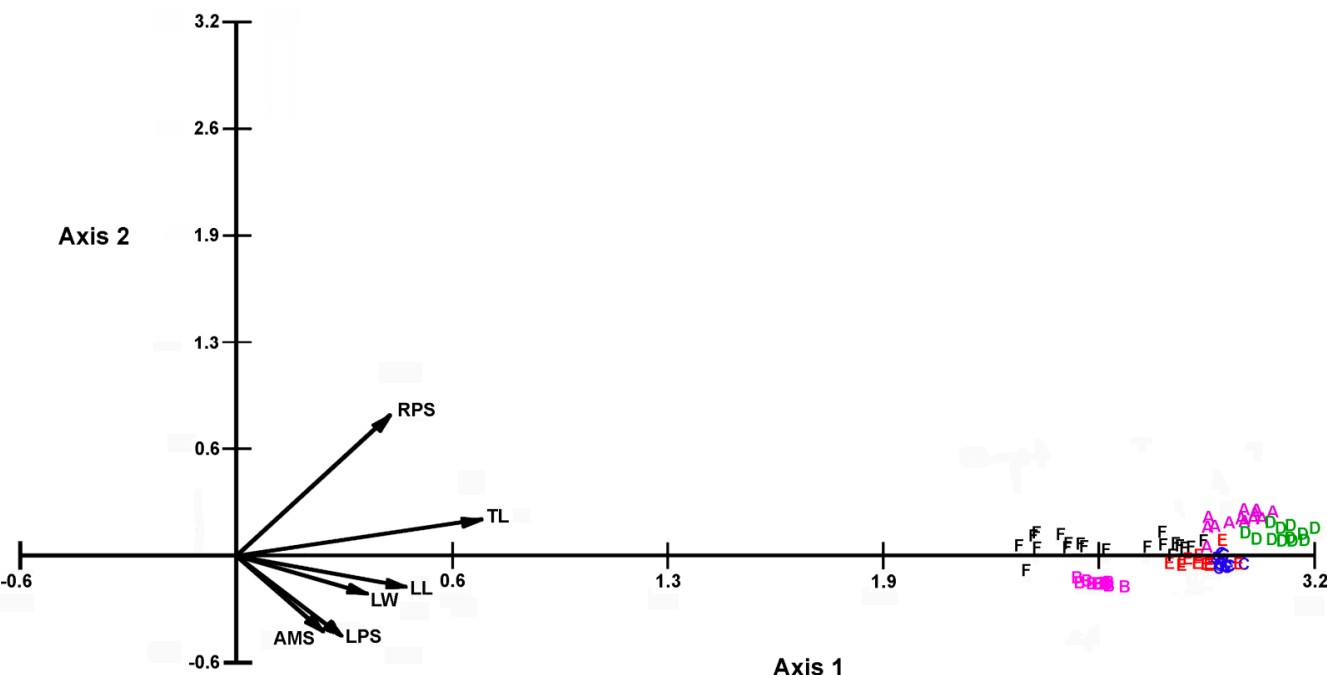

**Figure 5.** PCA biplot of *Keratella* populations analyzed. Symbols are as follow: **A** = ESU2, **B** = ESU3, **C** = ESU4, **D** = ESU1, **E** = ESU5 and **F** = ESU6. LT = Total lenght, LL = Lorica lenght, LW = Lorica width, RPS = Right posterior spine, LPS = Left posterior spine, AMS = Antero median spine.

### 3.3. Taxonomy

*Keratella tropica* species is diagnosed by a stiff lorica, bloated laterally. With six spines over the anterior dorsal margin. The dorsal lorica has a row of five median fields, four of these fields are hexagonal and the posteromedian remnant is smaller and square. Dorsal lorica is ornamented by a granular pattern. Posterior end of lorica slightly rounded, with two stiff posterior spines. In this species, the length of the posterior spines varies widely, with the right one longer than the left one. Although there are specimens where the left posterior spine is greatly reduced or absent.

The morphology of specimens from Yuriria and Tepatitlan-Yahualica populations compared with the descriptions in taxonomic keys and a careful review of the type specimens mounted in a permanent slide and deposited by [21] in the "Vermes" collection of the Zoological Museum of Berlin with catalog number 10,121 allowed us the conclusion that specimens ITS1-ESUII (ESU6 from COI) from Yuriria and Tepatitlan-Yahualica correspond to *Keratella tropica* s. str (See Figures 6 and 7).

Below, we present the taxonomic description of the three new *Keratella* species. In general these three new species present 19 fields over their dorsal plate: five median fields, four pairs of large polygonal lateral fields and three pairs of triangular marginal fields, all of them delimited by ridges. Moreover, we compared our specimens from these three new species with the type specimens and we confirm that these new species are different from *K. tropica* s. str.

Phylum Rotifera Cuvier, 1817
Class Eurotatoria De Ridder, 1957
Subclass Monogononta Plate, 1889
Order Ploima Hudson and Gosse, 1886
Family Brachionidae Ehrenberg, 1838
Genus *Keratella* Bory de St. Vincent, 1822
*Keratella cuitzeiensis* sp. nov.
Zoobank ID: urn:lsid:zoobank.org:act:7F194E23-7A36-46C4-90D9-EA38EB5F4380

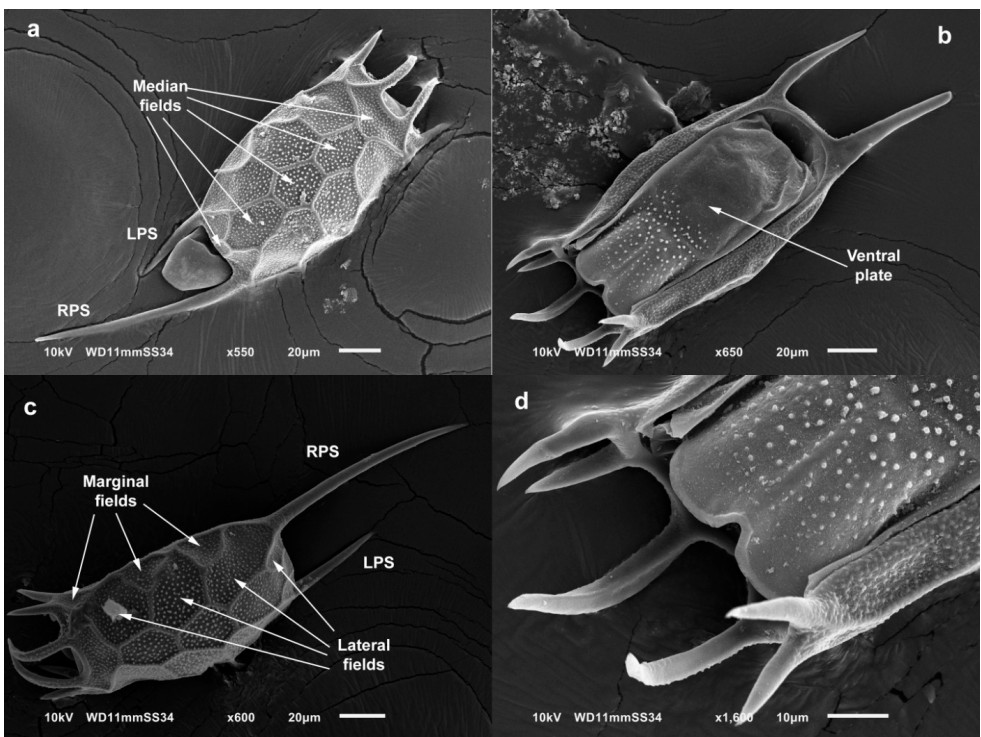

**Figure 6.** SEM photographs of *Keratella tropica* from Yuriria dam. (**a**). Dorsal view: arrows indicate the median fields, RPS (Right Posterior Spine), LPS (Left Posterior Spine). (**b**). Ventral view showing ornamentation over anterior part of the lorica. (**c**). Dorsal view: arrows indicate the lateral and the marginal fields, RPS (Right Posterior Spine), LPS (Left Posterior Spine). (**d**). Ventral view of the anterior part of the lorica.

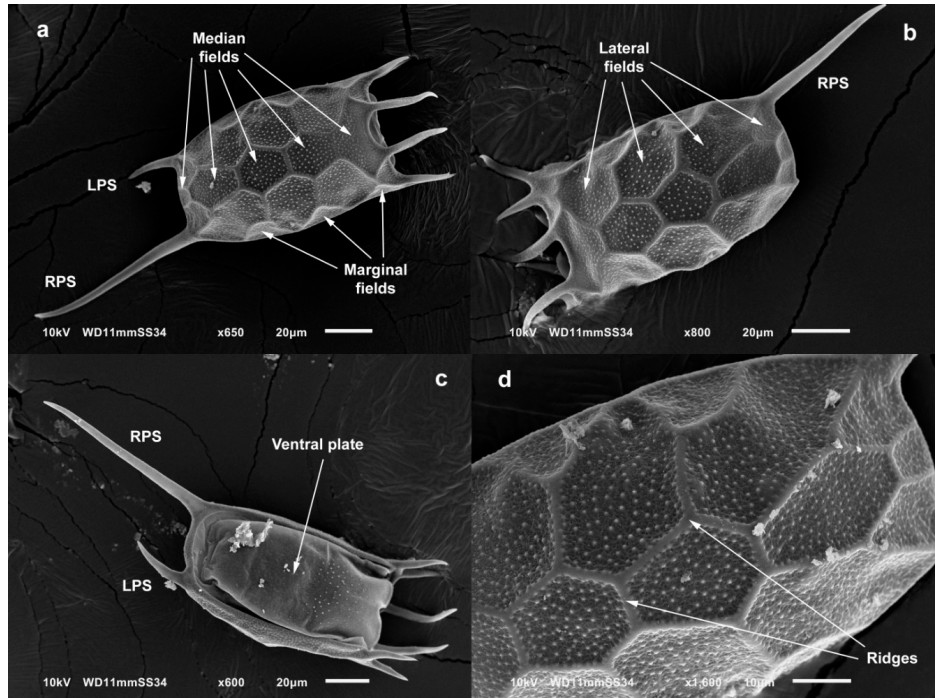

**Figure 7.** SEM photographs of *Keratella tropica* from Tepatitlan-Yahualica pond. (**a**). Dorsal view: arrows indicate the median and the marginal fields. (**b**). Dorsal view: arrows indicate lateral fields. (**c**). Ventral view, RPS (Right Posterior Spine), LPS (Left Posterior Spine). (**d**). Dorsal view closeup, arrows indicate a ridge.

Figures 8a,b and 9a,b.

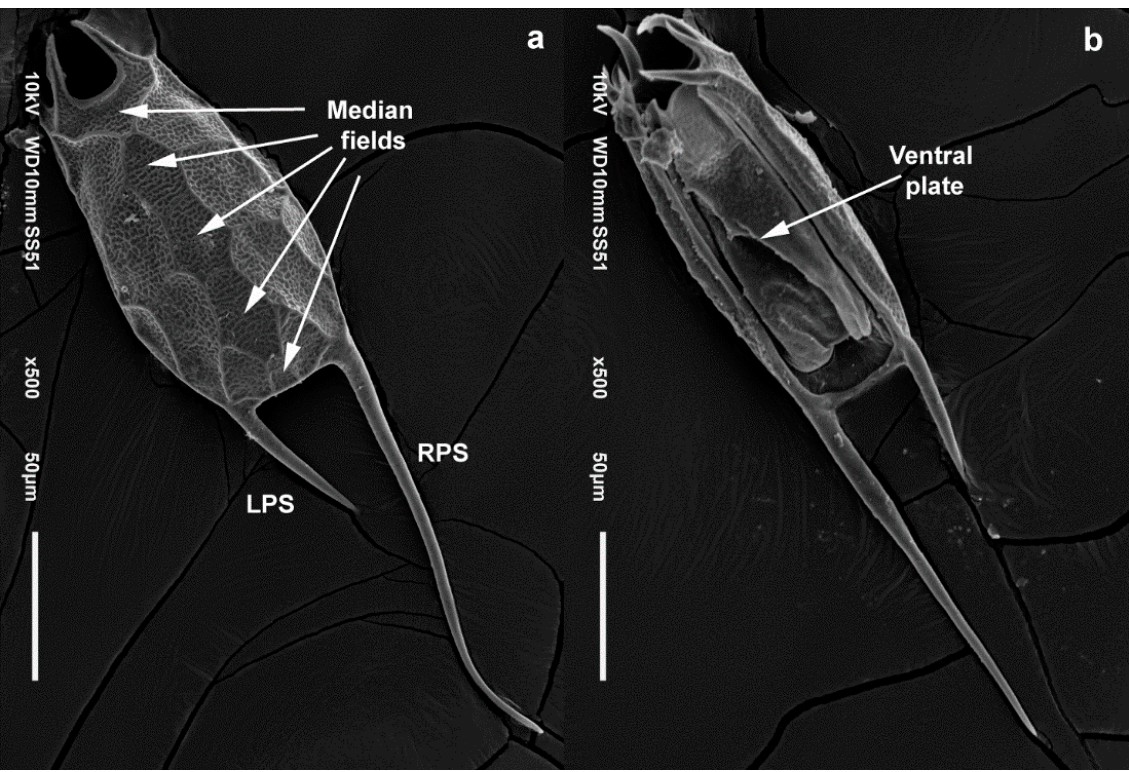

**Figure 8.** Morphology of *Keratella cuitzeiensis* sp. nov. (**a**). Dorsal view: arrows indicate the five median fields, RPS (Right Posterior Spine), LPS (Left Posterior Spine). (**b**). Ventral view. Females from Cuitzeo lagoon.

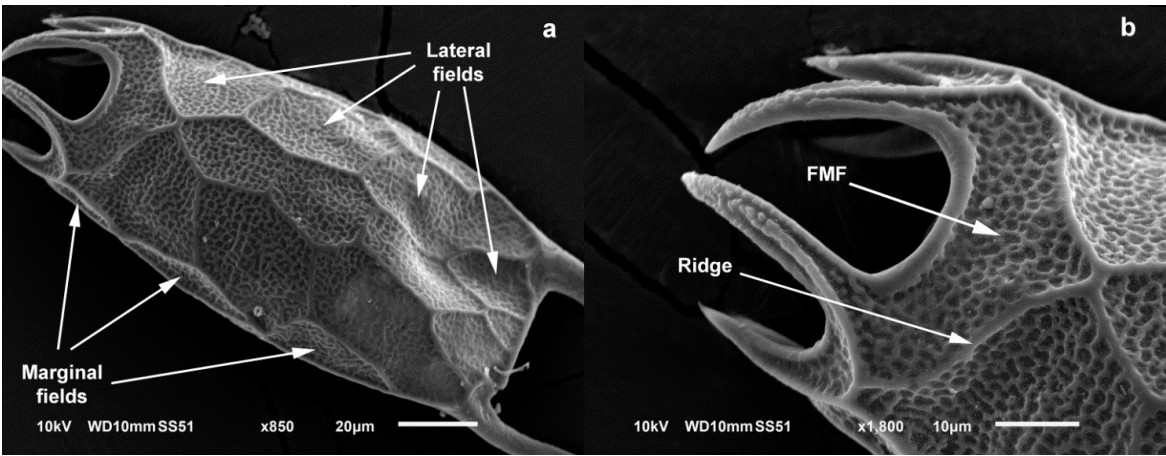

**Figure 9.** Morphology of *Keratella cuitzeiensis* sp. nov. (**a**). Dorsal view: arrows indicate the lateral and the marginal fields. (**b**). Anterior margin of the lorica in dorsal view, FMF (Frontomedian field), thin arrow indicate a ridge. Female from Cuitzeo lagoon.

Type locality and types. Cuitzeo lagoon is located in Michoacan, Mexico (19.8894 N, −100.9439 W). Sample collected on 25 February 2014. It is a shallow and saline water body, located at 1836 masl. The surface of the lagoon is 42,000 ha, with a maximum depth of 2.2 m.

Holotype: A parthenogenetic female mounted on a permanent slide. Paratypes: 10 females in a tube with ethanol. Holotype and paratypes are deposited in the Zooplankton Reference Collection of El Colegio de la Frontera Sur with accession numbers ECO-CH-Z-10589 and ECO-CH-Z-10590, respectively.

Differential diagnosis: *Keratella cuitzeiensis* sp. nov., most closely resembles the *Keratella tropica* species, *K. huapanguensis* sp. nov., and *K. albertae* sp. nov. It is diagnosed by a lorica that is a little bloated laterally. The anteromedian field is pentagonal. Mesomedian and posteromedian fields are hexagonal and elongated. The posteromedian remnant is conspicuous and slightly elongated. The dorsal plate is ornamented by a reticulate pattern. With two stiff posterior spines, the right one is longer than the left one (Figure 8a,b).

Description: Lorica stiff, the posterior margin of the lorica is slightly wider than the anterior margin. The posterior end of the lorica is almost straight, with two stiff and unequal posterior spines (Figure 8a). With three pairs of anterior spines that are short (Figure 9b). Anteromedian spines are recurved inward. Frontomedian field is an open short pentagon with lateral ridges prolonged into anteromedian spines (Figure 9a,b). The ventral plate is delicate, narrower and shorter than the dorsal plate. The ventral plate is bilobate, smooth without ornamentation in its anterior part (Figure 8b). Average measurements: 293.4 μm of total length, 75.4 μm of lorica width, 132.3 μm of lorica length, 134.8 μm of the right posterior spine, 49.7 μm of the left posterior spine (See Table 1).

Ecology and distribution: In Cuitzeo lagoon the new species coexists with *Brachionus quadridentatus* Hermann, 1783, *Cyclops* (Müller, 178), *Mastigodiaptomus patzcuarensis* (Kiefer, 1938), fishes (e.g., *Chirostoma* Swainson, 1839; *Xenotoca* Hubbs and Turner, 1939; *Zoogoneticus* Meek, 1902) and ostracods (*Potamocypris* Brady, 1870). The lagoon is a turbid environment with an average conductivity of 6595 μS/cm, temperature of 22 °C and pH 8 to 11.5. Cuitzeo is located in a region with a dry climate, with annual precipitation of between 6.0–150 mm. Specimens from this new species were also found in the Ignacio Ramirez dam and Timilpan pond located in the state of Mexico.

Etymology: The species name refers to the type locality where it was collected.

G + C content: ITS1 marker 0.293; COI gene 0.365.

*Keratella huapanguensis* sp. nov.

Zoobank ID: urn:lsid:zoobank.org:act:FD30C7A8-C63E-435D-A1C6-99003007789B

Figure 10a–d.

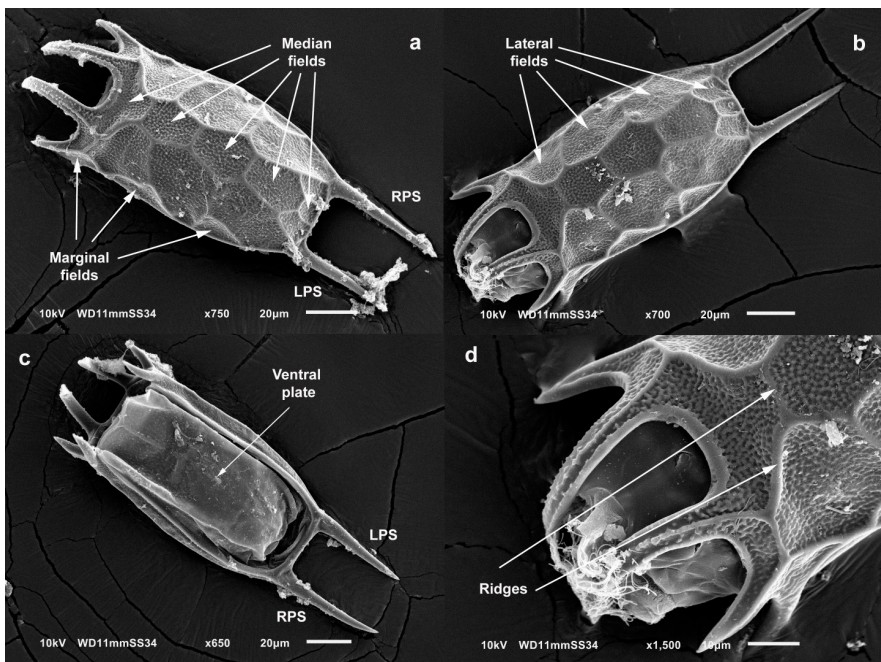

**Figure 10.** Morphology of *Keratella huapanguensis* sp. nov. (**a**). Dorsal view: arrows indicate the median and the marginal fields, RPS (Right Posterior Spine), LPS (Left Posterior Spine). (**b**). Dorsal view: arrows indicate lateral fields. (**c**). Ventral view, RPS (Right Posterior Spine), LPS (Left Posterior Spine). (**d**). Anterior margin of the lorica in dorsal view, arrows indicate a ridge. Females from Huapango dam.

Type locality and types: Huapango dam is located in the state of Mexico, Mexico (19.9483 N, −99.7144 W). Sample collected on 17 August 2014. It is a freshwater system, located at 2619 masl. The surface of the dam is 1000 ha, with a maximum depth of 14 m. It belongs to the basin of the Lerma River that runs 708 km from the state of Mexico to Jalisco, flowing out into Chapala Lake.

Holotype: A parthenogenetic female mounted on a permanent slide. Paratypes: 10 females in a tube with ethanol. Holotype and paratypes are deposited in the Zooplankton Reference Collection of El Colegio de la Frontera Sur with accession numbers ECO-CH-Z-10591 and ECO-CH-Z-10592, respectively.

Differential diagnosis: *Keratella huapanguensis* sp. nov. Most closely resembles the *Keratella tropica* species, *K. cuitzeiensis* sp. nov., and *K. albertae* sp. nov. It is diagnosed by a lorica almost rectangular in dorsal view. Anteromedian, mesomedian and posteromedian fields are hexagonal with almost the same size (Figure 10a,b). The posteromedian remnant is rounded and small (Figure 10a). It presents two short posterior spines (Figure 10a,c). The dorsal plate is ornamented by a reticular-granular pattern (Figure 10d).

Description: Lorica stiff, the anterior margin of the lorica is slightly wider than the posterior margin. The posterior end of the lorica is almost straight, with two stiff and relatively short posterior spines, the right one slightly longer than the left one (Figure 10a,b). With three pairs of anterior spines, which are elongated (Figure 10d). Anteromedian spines are the longest and recurved. Frontomedian field is an open short hexagon with ridges prolonged into anteromedian spines (Figure 10a,d). The ventral plate is delicate, narrower and shorter than the dorsal plate. The ventral plate is bilobate, smooth with some granules in its anterior part (Figure 10c). Average measurements: 201.3 μm of total length, 65.9 μm of lorica width, 113.3 μm of lorica length, 56.1 μm of the right posterior spine, 45.9 μm of the left posterior spine (See Table 1).

Ecology and distribution: In the Huapango dam the new species coexists with fish, e.g., *Girardinichthys multiradiatus* (Meek, 1904), "ajolote" *Ambystoma granulosum* Taylor, 1944, and "acocil" *Cambarellus montezumae* (Saussure, 1857). The sample was taken in the littoral zone among aquatic vegetation, and the water temperature and depth in that zone were 26 °C and 0.3 m respectively. Huapango is located in a region with a temperate sub-humid climate, with annual precipitation of between 700–1200 mm.

Etymology: The species name refers to the type locality where it was collected.

G + C content: ITS1 marker 0.296; COI gene 0.353.

*Keratella albertae* sp. nov.

Zoobank ID: urn:lsid:zoobank.org:act:DCA4000E-4B2B-4221-A15A-8CF15EF69B30

Figure 11a–d.

Type locality and types: Santa Teresa is a dam located in the state of Michoacan, Mexico (19.8886 N, −100.1722 W). The sample was collected on 17 August 2014. It is a freshwater system, located at 2307 masl. The surface of the lagoon is 149 ha, with a maximum depth of 49 m.

Holotype: A parthenogenetic female mounted on a permanent slide. Paratypes: 10 females in a tube with ethanol. Holotype and paratypes are deposited in the Zooplankton Reference Collection of El Colegio de la Frontera Sur with accession numbers ECO-CH-Z-10593 and ECO-CH-Z-10594, respectively.

Differential diagnosis: *Keratella albertae* sp. nov. Most closely resembles the *Keratella tropica* species, *K. cuitzeiensis* sp. nov., and *K. huapanguensis* sp. nov. It is diagnosed by a lorica slightly bloated in dorsal view. Anteromedian, mesomedian and posteromedian fields are hexagonal, of which posteromedian field is more elongated (Figure 11a,b). The posteromedian remnant is a conspicuous and elongated field (Figure 11b). The dorsal plate is ornamented by a granular pattern (Figure 11d). With two stiff posterior spines, the right one is longer than the left one (Figure 11a).

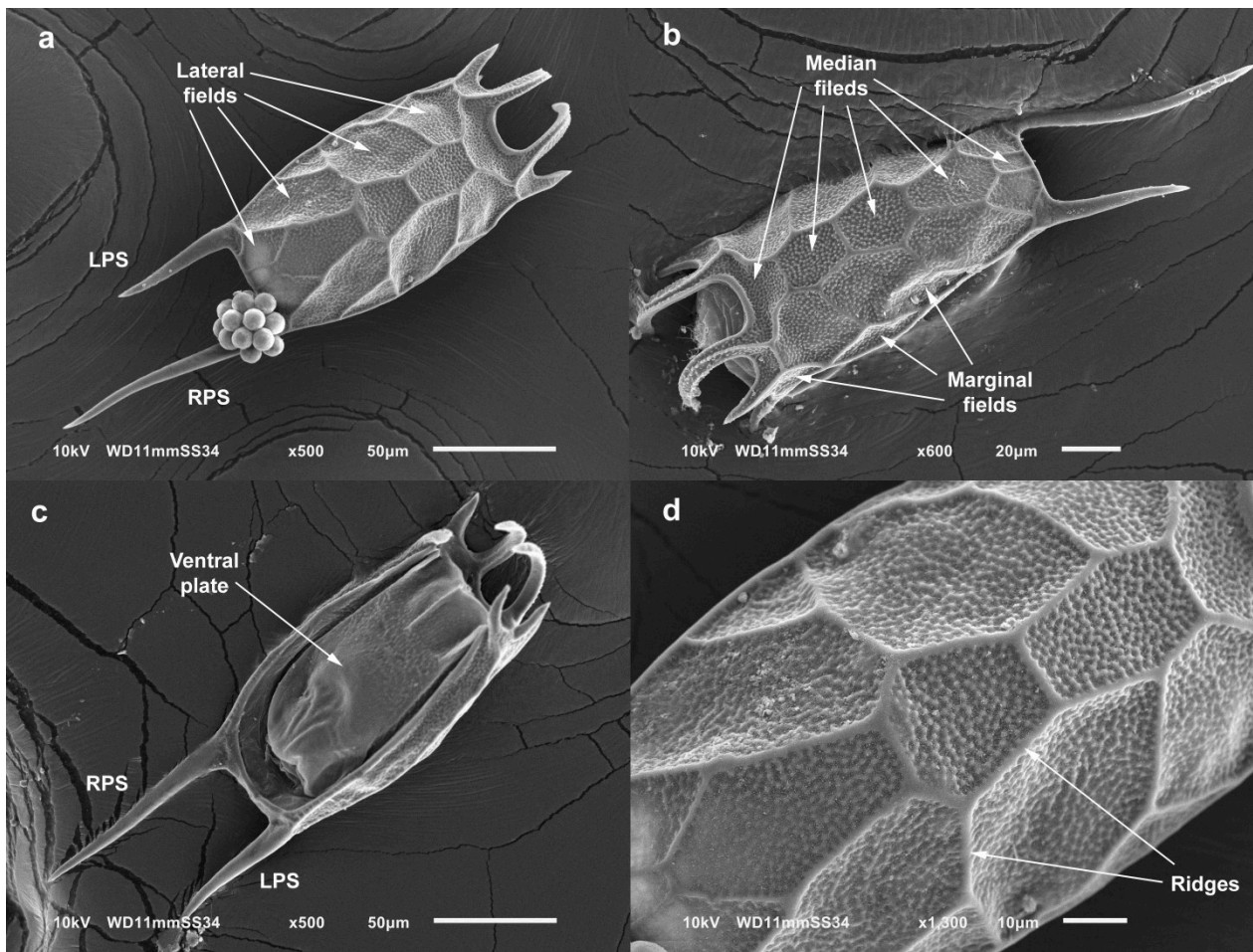

**Figure 11.** Morphology of *Keratella albertae* sp. nov. (**a**). Dorsal view: arrows indicate the lateral fields, RPS (Right Posterior Spine), LPS (Left Posterior Spine). (**b**). Dorsal view: arrows indicate the median and the marginal fields. (**c**). Ventral view. (**d**). Dorsal view closeup, arrows indicate a ridge. Females from Santa Teresa dam.

Description: Lorica stiff, the anterior margin of the lorica is slightly wider than the posterior margin. The posterior end of the lorica is almost straight, with two stiff and unequal posterior spines, the right one longer than the left one (Figure 11a,c). With three pairs of anterior spines, which are elongated. Anteromedian spines are the longest and recurved (Figure 11a,c). Frontomedian field is an open short hexagon with ridges prolonged into anteromedian spines (Figure 11a,b). The ventral plate is delicate, narrower and shorter than the dorsal plate. The ventral plate is bilobate, smooth with some ornamentation in its anterior part (Figure 11c). Average measurements: 265.6 μm of total length, 80.6 μm of lorica width, 128.4 μm of lorica length, 101.4 μm of the right posterior spine, 63 μm of left posterior spine (See Table 1).

Ecology and distribution: In Santa Teresa dam the new species coexists with the common carp (*Cyprinus carpio* Linnaeus, 1758). The sample was taken in the littoral zone, and the water temperature in that zone was 26 °C. Santa Teresa is located in a region with a temperate sub-humid climate, with annual precipitation of between 700–1000 mm. Specimens from this new species were also found in the Kohunlich pond located in the state of Quintana Roo.

Etymology: The species name refers to the name of the mother of AEGM.

G + C content: ITS1 marker 0.305; COI gene 0.349.

## 4. Discussion

Cryptic species are by definition, a set of closely related species that share very similar morphological features, and therefore are not readily distinguished [2]. Additionally, the tiny size (50–2000 microns), and translucent body of the rotifers make sometimes difficult to observe the morphological characteristics of the specimens under a compound microscope. For all above the specimens can be confused with morphologically similar species and be mistakenly identified. For example, due to the morphological stasis of the external features that characterize the *B. plicatilis* species complex [8,29], the use of molecular analysis was a fundamental basis to help unravel the cryptic diversity within this group [2]. Our study demonstrates that taxonomy-based only on morphology is not effective at providing an accurate assessment of the diversity of the *Keratella* taxa. Combining morphology with other data as genetics and ecology has demonstrated to be a more reliable approach to study diversity in rotifers and new species have been described as is the case of *Brachionus paranguensis* from central Mexico [9]. In the present study, DNA taxonomy through the use of the two markers COI and ITS1 represented an important tool that helped to confer identity to the new entities, named here as *Keratella cuitzeiensis* sp. nov., *Keratella huapanguensis* sp. nov., and *Keratella albertae* sp. nov.

Our results suggest that the divergence between the COI and ITS1 is ancient. The sequence divergence between the three new species found in both mitochondrial and nuclear markers (4–20% for COI and 3–10% for ITS1) exceeds the values usually found between congeneric species, indicating that each of these species has an independent evolutionary history. These genetic divergences are similar to the values found in species complex as *Brachionus plicatilis* (11–23% for COI and 2.5–22% for ITS1, [2]) and *B. calyciflorus* (9–13% for COI and 3–6% for ITS1, [11]).

However, we found mitonuclear discordance between the mitochondrial COI and the nuclear ITS1, as the nuclear marker revealed four ESUs (ESUI-ESUIV) and mitochondrial six ESUs (ESU1-ESU6). Mitonuclear discordance between COI and ITS1 in rotifers has already been observed in previous studies with *B. calyciflorus* [51], *K. cochlearis* and *Polyarthra dolichoptera* [49], and *B. paranguensis* by [9]. In fact, the discordance observed here is similar to the latter species, where ITS1 did not split the species as COI. Mitonuclear discrepancies were attributed to processes such as hybridization, incomplete lineage sorting and horizontal gene transfer [7,49]. Nevertheless, other factors as concerted evolution of the ribosomal nuclear markers can be considered to explain the incongruence between the mitochondrial and nuclear phylogenies [52].

Concerted evolution is defined as the coordinate evolution of repetitive DNA sequences (such as rDNA) resulting in a sequence similarity of repeating units that are greater within than among species [53]. Therefore, concerted evolution leads to sequence homogeneity within a species, but also a divergence between species [54]. COI gene evolves much faster than the nuclear ITS1 marker [9], therefore the rapid mutational rate of COI explains the high genetic variation found for this gene within the examined *Keratella* taxa. Nevertheless, it is hard to say what process is responsible for the mitonuclear discordance found in our study. We speculate that incomplete lineage sorting can be the most likely cause of the mitonuclear discordance observed in our work. This is because differentiation has not been completed in the nuclear marker. However, other processes as concerted evolution or hybridization can be operating. Further research will be needed to determine which process is responsible for the mitonuclear discordance observed in the *Keratella* taxa examined in the present study.

### 4.1. Morphology

*Keratella tropica* is a cosmopolitan species that shows morphological variation in the length of the lorica and posterior spines [55,56]. However, with the morphological and morphometric analyses, we could observe morphological differences between our *K. tropica* specimens and specimens of the three new species that can allow us distinguishing among them. Differences were observed in the shape of the median fields, length of the

posterior and anterior spines, dorsal lorica ornamentation, body size and lorica shape. Other differences were observed in the ecological preferences of the species (see next section).

### 4.2. Distribution and Ecological Comments

*Keratella tropica* is distributed in tropical and subtropical regions of the world [23,56,57]. In America, it has been recorded in temperate zones from Mexico, USA and Argentina (Patagonia) [58–60]. The three new species are distributed in central Mexico, which is a temperate region with mountains.

Although we did not measure environmental parameters (except temperature), we want to make some remarks about the environment of the water bodies of the three new species based on published literature. This is important because differences in salinity preference among *Brachionus* species, for example, is considering another factor that has allowed discrimination between species [9]. The water bodies where these three new species were found show some differences in their water chemistry [61–64]. Cuitzeo lagoon is the most saline water body displaying salt levels up to 5 g L$^{-1}$ [65], with pH fluctuating between 9.8 and 10.4, and conductivity of 6595 μS cm$^{-1}$ [61]. Santa Teresa dam displays levels of conductivity between 200–280 μS cm$^{-1}$, and alkalinity of 120 mg L$^{-1}$ [63]. Whereas, Huapango dam displays levels of alkalinity between 24 to 49 mg L$^{-1}$, with pH fluctuating among 5.7 to 7.6 and dissolved oxygen among 5.6 to 9.4 mg L$^{-1}$ [62]. Therefore, the Cuitzeo lagoon is the system that has the most particular environmental conditions from all systems.

Moreover, Cuitzeo lagoon is also an interesting case, because is an antique lagoon originated during the upper Miocene and experimented several changes during the Pleistocene [66]. Today Cuitzeo is a saline system affected by anthropogenic activities. However, it is an important water body due that harbors some native fish species, for example, *Chirostoma compressum* de Buen, 1940; *C. jordani* Woolman, 1894 and *Xenotoca variata* (Bean, 1887) [67], and recently a copepod species *Mastigodiaptomus patzcuarensis* was found here [68]. Recently, another cryptic species belonging to the *Brachionus quadridentatus* species complex was found in this lagoon [7].

In addition, it was shown that certain environmental parameters as salinity and temperature of the water have important influences over the adaptation of the species and eventually over their genetic differentiation [5,69]. Some sibling species within the *B. plicatilis* species complex possess differential responses to salinity. For example, ref. [70] found that *B. plicatilis* s. str. occurs at low to high salinities (3–45 g L$^{-1}$), whereas *B. ibericus* Ciros-Pérez, Gómez and Serra, 2001, and *B. rotundiformis* Tschugunoff, 1921 occur in waters with medium to high salinities (8–50 and 10–57g L$^{-1}$ respectively). However, *B. ibericus* occurs at high temperatures (>15 °C) and *B. rotundiformis* at temperatures between 10–30 °C. A laboratory study with *B. manjavacas* Fontaneto, Giordani, Melone and Serra, 2007 presented similar results where the optimal salinity for this species was observed in the range 10–30 g L$^{-1}$ [71]. Whereas [72] reported levels of optimal salinity for *B. asplanchnoides* Charin, 1947 between 3.8–8.5 g L$^{-1}$. However, *B. paranguensis* a species recently described seems to be adapted to high salinity (>25 g L$^{-1}$), which allowed the species to colonize the hypersaline volcanic maar lake Rincón de Parangueo [9]. Maar lake Rincón de Parangueo is 66 km away from the Cuitzeo lagoon. According to the above, differences in salinity tolerance can be considered as an additional parameter to discriminate the species in some brachionids.

## 5. Conclusions

This is the first study conducted on several populations of *Keratella* from Mexico using integrative taxonomy. A formal description was provided for *Keratella cuitzeiensis* sp. nov., *Keratella huapanguensis* sp. nov., and *Keratella albertae* sp. nov., combining morphology and genetics. These three new species are related to *K. tropica* species. Comparison of SEM images and morphometry among the three new species showed differences in body

shape, the shape of median fields, length of the posterior and anterior spines, body size, and dorsal lorica ornamentation. Therefore, DNA sequences along with morphological data support the existence of the three new species. Results of genetic variation were different among the two markers used, with a higher genetic divergence in the COI gene (six ESUs) compared to the ITS1 marker (four ESUs), and thus, providing evidence of mitonuclear discordance. This incongruence might be due to differences in mutation rate between markers, probably because of incomplete lineage sorting. Environmental conditions reported for the water systems of the three new species suggest different salinity preferences of the species, with *Keratella cuitzeiensis* sp. nov., adapted to a more saline water body than *Keratella huapanguensis* sp. nov., and *Keratella albertae* sp. nov.

**Supplementary Materials:** The following are available online at https://www.mdpi.com/article/10.3390/d13120676/s1, Table S1: Geographic coordinates of the sampling sites, Table S2: PCR profile for ITS1 marker, Table S3: GenBank accessions of COI and ITS1 used in this study, Table S4: Percentages of uncorrected genetic distances of COI-ESUs, Table S5: Percentages of uncorrected distances of ITS1-ESUs, Table S6: Measurements of *Keratella* specimens.

**Author Contributions:** Conceptualization, A.E.G.-M.; validation, A.E.G.-M., O.D.-D. and M.E.-G.; formal analysis, A.E.G.-M., O.D.-D. and M.E.-G.; resources, O.D.-D. and M.E.-G.; data curation, A.E.G.-M.; writing—original draft preparation, A.E.G.-M.; writing—review and editing, A.E.G.-M., O.D.-D. and M.E.-G.; funding acquisition, O.D.-D. All authors have read and agreed to the published version of the manuscript.

**Funding:** This research was funded in part by grant number CIC-UMSNH-2015.

**Institutional Review Board Statement:** Not applicable.

**Data Availability Statement:** The data presented in this study are available in Supplementary Materials; Tables S1–S6.

**Acknowledgments:** We thank Birger Neuhaus, Scientific Head Collection "Vermes" of the Zoological Museum of Berlin, Germany, who sent us several photographs of the *Keratella tropica* types deposited by Apstein (1907). We thank Adrian Cervantes Martínez, who kindly assisted us with the PCA analysis.

**Conflicts of Interest:** The authors declare no conflict of interest.

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
