# Peer review of "Uncovering Hidden Diversity: Three New Species of the Keratella Genus (Rotifera, Monogononta, Brachionidae) of High Altitude Water Systems from Central Mexico†"

_diversity, doi:10.3390/d13120676_

Round 1

Reviewer 1 Report

1. Have a native English speaker read through the entire article and correct the small grammatical errors (there are quite a few of them).

2. Add Rotifera to the title:

Uncovering Hidden Diversity: three new species of the Keratella genus (Rotifera, Monogononta, Brachionidae) of high altitude water systems from Central Mexico

3. On page 4, line 123, clearly differentiate between the mitochondrial and nuclear sequences. Many general readers may not catch the difference and it is a good idea to remind them of the differences at this point. Clearly differentiating these can be done like this: “We downloaded COI (mitochondrial) and ITSI (nuclear) sequences …..”

4. In the Results section, besides Figures 2 and 3, add the following trees:

4a. For each new species, zoom in on the relevant portions of the mitochondrial and nuclear trees and clearly highlight where the new species are. Figures 2 and 3 are very confusing in terms of trying to figure out where each new species is located on the tree.

4b. Add a morphological tree which includes, at minimum, Keratella tropica 1 and 2 as well as each of the new species.

Author Response

Response to Reviewer 1 Comments

Point 1: Have a native English speaker read through the entire article and correct the small grammatical errors (there are quite a few of them).

Response 1: We corrected the grammatical errors.

Point 2: Add Rotifera to the title:

Uncovering Hidden Diversity: three new species of the Keratella genus (Rotifera, Monogononta, Brachionidae) of high altitude water systems from Central Mexico.

Response 2: We added Rotifera to the title.

Point 3: On page 4, line 123, clearly differentiate between the mitochondrial and nuclear sequences. Many general readers may not catch the difference and it is a good idea to remind them of the differences at this point. Clearly differentiating these can be done like this: “We downloaded COI (mitochondrial) and ITSI (nuclear) sequences …..”

Response 3: We added (mitochondrial) and (nuclear) to the text.

Point 4: In the Results section, besides Figures 2 and 3, add the following trees:

Point 4a: For each new species, zoom in on the relevant portions of the mitochondrial and nuclear trees and clearly highlight where the new species are. Figures 2 and 3 are very confusing in terms of trying to figure out where each new species is located on the tree.

Response 4a: We corrected figures increasing the font size in order to make more visible the text on the phylogenetic trees for COI and ITS.

Point 4b: Add a morphological tree which includes, at minimum, Keratella tropica 1 and 2 as well as each of the new species.

Response 4b: We did not include a morphological tree because this is more appropiated for a phylogeny and our work is focused to describe new species. We only construted phylogenetic trees in order to show graphically that the new species are independent entities related to the other Keratella species, but our work is not a phylogenetic study. Morphology of K. tropica and the new species is described in the manuscript.

Reviewer 2 Report

The text is clear and well articulated, the methods are appropriate, the morphological and molecular differences between the species are supported, the discussion is adequate to the results. The paper can be accepted as it is.

Author Response

Response to Reviewer 2 Comments

Point 1: The text is clear and well articulated, the methods are appropriate, the morphological and molecular differences between the species are supported, the discussion is adequate to the results. The paper can be accepted as it is.

Response 1: OK

Reviewer 3 Report

This is a very interesting species dealing with species delimitation in monogonont rotifers, more specifically within K. tropica complex.

I’ve made all my comments and suggestion within the PDF of the presented MS.

Major issues I noted during the evaluation was:

  1. Bad quality and low visibility in vector graphics present in the MS (phylogenetic trees and PCA plot). I made some suggestion about this in the file.
  2. As for me the differential diagnosis for new taxa is not complete. The authors differentiate new species with K. tropica only whereas all of them are similar to each other. I strongly recommend to use in each differential diagnosis also other species newly described in this study.

Round 2

Reviewer 1 Report

The manuscript makes a valuable contribution to our understanding of Rotifera diversity and cryptic species.

Minor suggestions/corrections:

Abstract:

Change: “The correct identification of the species…” to “The correct identification of species is..” (remove the before species).

Line 605: change “to” before Keratella to “the” (… the Keratella...)

Line 716: change “to” before Keratella to “the” (… the Keratella...)

Line 843: change “to” before Keratella to “the” (… the Keratella...)

Author Response

Response to Reviewer 1 Comments

Abstract:

Change: “The correct identification of the species…” to “The correct identification of species is..” (remove the before species).

Response 1: We corrected with “The correct identification of species is.. 

Line 605: change “to” before Keratella to “the” (… the Keratella...)

Response 2: We changed with “the”

Line 716: change “to” before Keratella to “the” (… the Keratella...)

Response 3: We changed with “the”

Line 843: change “to” before Keratella to “the” (… the Keratella...)

Response 4: We changed with “the”